# TINY-STYLEWIZARD: UNLEASHING THE POTENTIAL OF SMALL LANGUAGE MODELS IN COMPLEX STYLE TRANSFER

## ABSTRACT

Text style transfer is a crucial task in natural language processing. While previous studies focused on simple styles like sentiment and formality, they overlooked the transfer of valuable complex styles. In this paper, we propose a framework named Tiny-StyleWizard to address this challenge. It first generates a specialized dataset retaining key aspects of the desired complex style based on diverse corpora and a large language model (LLM) and then fine-tines a small language model to achieve the goal of complex style transfer. Additionally, a novel evaluation protocol is devised to rank the quality of the generated specialized dataset and to measure the performance of different models. Extensive experiments on two representative complex style transfer tasks reveal that small language models like BART-base/large can produce stylized text on par with ChatGPT while the tinier ones like T5-mini (about 30M parameters) could surpass the state-of-the-art models. Intriguingly, Our investigation on the efficient construction of the training corpus shows the phenomenon named "less is more" and the subsequent observation similar to "ungrokking", emphasizing the supreme importance of data quality. Further exploration also showcases the sufficient diversity of the generation texts obtained by our Tiny-StyleWizard framework.

## 1 INTRODUCTION

Numerous tasks exist within the realm of natural language processing (NLP), and text style transfer has emerged as one of the focal points of interest Li et al. (2016); Pryzant et al. (2020); Santos et al. (2018). The goal is to create a variety of texts that maintain their original meaning with the targeted style. Recently, pre-trained language models (PLMs), such as ChatGPT, GPT-4 OpenAI (2023) or LLaMA Touvron et al. (2023a;b), have significantly propelled the field of text style transfer. These models, after being pre-trained on extensive text collections and fine-tuned with style-specific data, have demonstrated impressive results.

Although many text style transfer studies concentrate on basic styles (e.g., sentiment or formality transfer) Dai et al. (2019a), there remains a considerable void in the area of complex text style transfer. One typical complex style is the personality, as illustrated in Figure 1. This task was initially proposed by BTTS Xu et al. (2023), which enhanced the universal style transfer model, TextSETTR Riley et al. (2020), through contrastive learning. Despite BTTS's ability to achieve cutting-edge results in complex style transfer tasks, it has notable limitations: it struggles with accurate control over the stylized text and may not consistently adhere to the desired style or original meaning. Moreover, these models often fall short compared to parallel methods, particularly in maintaining semantic integrity. We argue that complex style transfer poses difficulties in terms of model, data, and evaluation, as it requires models to not only understand the nuances of complex styles but also to integrate and harmonize components from every facet of complex styles. The challenges are:

**Challenge 1: Model Size Barrier in Achieving Efficient Transfer**. Large language models (LLMs) show promise in complex text style transfer tasks, but their size and closed-source nature pose challenges Wei et al. (2022); Xu et al. (2023). Our research is investigating if smaller models like BART or T5 can offer comparable performance Lewis et al. (2020); Raffel et al. (2020).

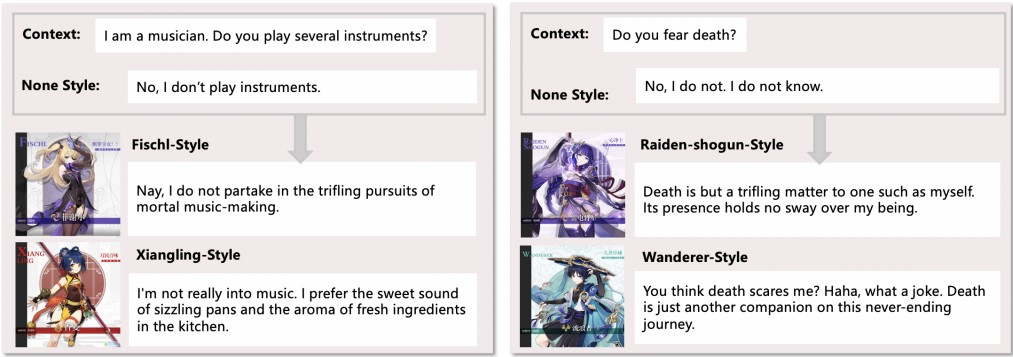

Figure 1: Two examples for personality transfer tasks. Given a context and recommended response with no style, the model is required to generate a personal response conforming to the persona information of the character. Fischl exhibits characteristics of chuunibyou and tsundere, often employing poetic language in her expressions. Xiangling is a highly skilled chef, renowned for her culinary expertise. The Raiden Shogun holds the position of the divine ruler of the Inazuma Shogunate, embodying the power of thunder and lightning. The Wanderer displays a tendency for arrogance and a dismissive attitude towards others.

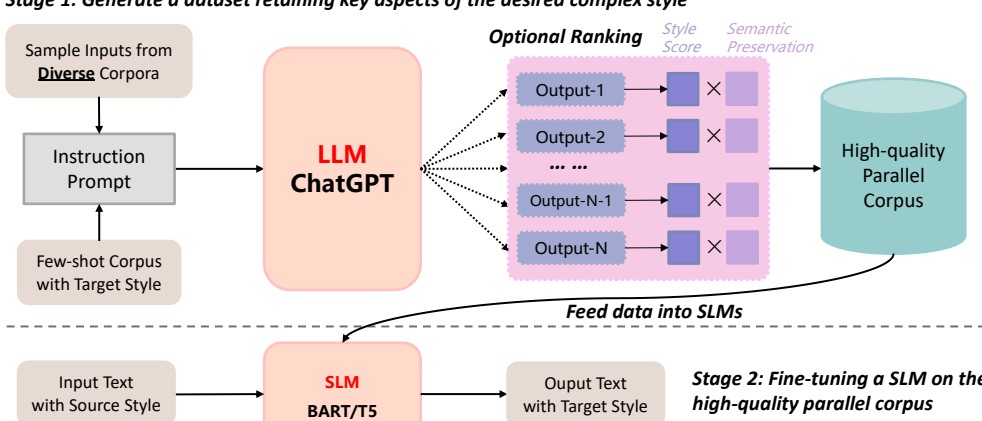

Figure 2: The Tiny-StyleWizard framework for complex text style transfer. It is composed of two stages. During the first stage, it generates a dataset retaining the key aspects of the desired complex style guided by in-context learning based on ChatGPT. During the second stage: we aim to unleash the capability of the small language model for complex style transfer by fine-tuning.

**Challenge 2: Limited Data and Generalization Hurdles**. Datasets for complex styles are often limited and require skilled writers, posing challenges for model fine-tuning. Acquiring sufficient annotated data for a broad range of styles in complex style transfer tasks is difficult. This constraint negatively impacts the generalization ability of the fine-tuned model.

**Challenge 3: Shortcomings of Traditional Metrics in Evaluation**. Traditional style classifiers and standard semantic metrics like BLEU Reif et al. (2021) and BERT-score Zhang et al. (2019) often fall short in handling complex styles and subtle stylistic changes, underscoring the need for enhanced measures in complex text style transfer. Moreover, the critical yet often neglected issue of text diversity in generated outputs also warrants attention.

**Solutions**. The limited ability of small language models in complex style transfer is unclear, with uncertainty whether it is due to the complexity of the task or the scope and diversity of training corpora. To investigate, we propose Tiny-StyleWizard (shown in Figure 2), a framework that generates a focused, smaller dataset retaining key aspects of the desired complex style, and explores the

potential of small language models through fine-tuning. Evidently, the creation of the specialized dataset addresses the issue of data scarcity (**Challenge 2**) while fine-tuning solves the issue of large model sizes (**Challenge 1**). As for **Challenge 3**, we devise a novel evaluation protocol. This protocol includes a few-shot ChatGPT-based style classifier for measuring style transfer accuracy and an enhanced LLM score metric for semantic preservation. This metric addresses the shortcomings of traditional metrics like BLEU Papineni et al. (2002) and BERT-score Zhang et al. (2019), which struggle with complex style transfer. In contrast to previous studies, we also evaluate text diversity using n-gram overlap.

To evaluate the efficacy of our proposed framework, we experiment on two representative complex style transfer tasks: personality transfer Xu et al. (2023) and narrative style transfer of long paragraphs Zhu et al. (2022). We conduct comprehensive experiments on models of various scales and types. We discover that pre-trained language models with hundreds of millions of parameters (like BART-base/large) can produce stylized text on par with ChatGPT. Subsequently, we aim to explore the construction of datasets that facilitate more efficient training of small language models for complex text style transfer, observing the classic **less is more** phenomenon and obtaining a subsequent observation similar to **ungrokking** Varma et al. (2023). Less is more refers to the phenomenon that models trained on a smaller but high-quality corpus outperform those trained on a larger corpus of moderate quality, while ungrokking refers to the observation that high-performing models experience a significant degradation in performance if they continue to train on mediocre or low-quality corpus. Moreover, we delve deeper into investigating the minimum size of models that can possess the capability of complex style transfer. Our investigation reveals that smaller language models (about 31M parameters) exhibit inferior performance compared to ChatGPT but outperform the state-of-the-art models for this task. Furthermore, the stylized text generated by our model demonstrates a notable degree of diversity, providing additional evidence for the practicality of the Tiny-StyleWizard framework.

## 2 TINY-STYLEWIZARD FRAMEWORK

Our framework called Tiny-StyleWizard (illustrated in Figure 2) consists of two stages. In the first stage, with the assistance of an LLM instantiated by ChatGPT (gpt-3.5-turbo-0315), it generates a diverse parallel corpus that retains the key aspects of the target complex style. During this process, we may filter and rank the quality of the generated corpus, despite the impressive text generation capabilities of LLMs. In the second stage, the framework fine-tunes a small-scale language model using the aforementioned generated parallel corpus to unleash the capability of these models for complex text style transfer. We will elaborate on these two stages and the specific implementation of corpus filtering and ranking in this section. We name the first stage as Dataset Generation while the second one as Fine-Tuning Small Language Models.

### 2.1 STAGE 1: DATASET GENERATION

As previously stated, the underlying principle of the Tiny-StyleWizard framework is to curate a corpus that amalgamates all the qualitative aspects inherent in the desired complex style, while simultaneously ensuring that the corpus is more focused and content-constrained. To actualize this, we lean heavily on the cutting-edge LLMs such as ChatGPT. The model is designed to generate copious amounts of synthetic content, guided by specific instructions. Our directive to these models is to generate content that mirrors the target complex style.

However, employing LLMs to produce training data presents a significant challenge: the creation of a dataset that boasts sufficient diversity Eldan & Li (2023). To surmount this hurdle, we adopt a simple but effective approach, sampling inputs from corpora that span a wide array of domains. In the context of the personality transfer task, where data is often in short supply, we turn to the DialogStudio corpora Zhang et al. (2023) for inputs. From this corpus, we randomly select 10,000 single-turn dialogues. These samples are then fed into ChatGPT to generate responses that align with the desired character personality. This strategy is not exclusive to the personality transfer task. It is also employed in tasks that involve narrative style transfer for extended paragraphs. For such tasks, we source inputs from corpora written by the same author.

Table 1: Different Metrics for Style Transfer Accuracy and Semantic Preservation.

| Dataset | Style Accuracy | | Semantic Preservation | | | | | |
| | Fine-tuned | ChatGPT | BLEU-1 | BLEU-2 | BS-P | BS-R | BS-F1 | LLM-score |
|---|---|---|---|---|---|---|---|---|
| Genshin | 31.20 | **88.72** | 15.17 | 8.16 | 11.03 | 31.49 | 21.03 | **86.44** |
| Shakespeare | **99.41** | 99.23 | 37.46 | 23.88 | 26.32 | 44.93 | 35.44 | **93.47** |
| LuXun | **99.60** | 97.67 | 30.71 | 10.75 | 33.54 | 43.33 | 38.47 | **94.54** |
| JinYong | **99.60** | 99.45 | 24.35 | 8.81 | 36.45 | 44.53 | 40.54 | **95.66** |

## 2.2 STAGE 2: FINE-TUNING SMALL LANGUAGE MODELS

With the synthetic parallel corpus at hand, we use the pre-trained seq2seq model to implement the small language models specialized for the magician capabilities of complex style transfer. For the English datasets, we consider different scales of BART Lewis et al. (2020) or T5 Raffel et al. (2020). For the Chinese ones, we consider the Chinese version of BART Shao et al. (2021) or LongLM Guan et al. (2021).

## 2.3 OPTIONAL RANKING: AUTOMATIC EVALUATION PROCEDURE

In order to filter and rank the quality of the dataset generated in Stage 1, we are inevitably faced with a fundamental question: what criteria should be used for filtering and ranking? Essentially, this question revolves around how to evaluate the quality of ChatGPT-generated corpora for complex text style transfer tasks. To address this question, it is important to examine how previous research has evaluated the effectiveness of text style transfer models. Historically, research in this domain has primarily focused on two aspects: style transfer accuracy and semantic preservation. Style transfer accuracy has been evaluated using fine-tuned Convolutional Neural Networks (CNN) Dai et al. (2019a) or BERT-based models Riley et al. (2020). Semantic preservation, on the other hand, has been assessed using metrics such as BLEU Reif et al. (2021) or BERT-score Zhang et al. (2019). Notably, recent studies by Zhu et al. (2022) and Dai et al. (2019a) have emphasized the tradeoff between content preservation and transfer accuracy. However, our analysis has identified certain limitations with existing metrics pertaining to style accuracy and semantic preservation. This analysis prompts us to propose enhancements. In our evaluation, we employ style transfer accuracy and semantic preservation as key metrics, while also introducing improvements to address the identified issues. The metrics we utilize, along with our proposed enhancements, are as follows:

**Metric 1: Style Transfer Accuracy**. Previous work Dai et al. (2019b); Zhu et al. (2022) usually trains a style classifier using the style-labeled corpus and regards the classifier score as the style accuracy. We argue that this is not applicable to complex style transfer where the style-specific corpus may be lacking and the performance may be sub-optimal. Instead, we use few-shot complex-style-specific exemplars to construct a classifier based on ChatGPT. To test the effectiveness of this approach, we conducted experiments on various datasets. The experimental results are shown in Table 1, illustrating that the ChatGPT-based classifiers have stronger or comparable classification capabilities for different styles than the counterparts trained via traditional fine-tuning.

**Metric 2: Content Preservation**. Previous studies Zhu et al. (2022) commonly employ metrics such as BLEU-$n$ ($n = 1, 2$) and BERT-score to measure the lexical and semantic similarity between generated texts and input texts. However, these metrics have limitations when evaluating semantic preservation in complex text style transfer tasks. In Table 1, we present the BLEU-$n$ and BERT-score calculations between input texts and stylized texts generated by ChatGPT. Manual inspection verifies that the ChatGPT-generated corpus preserves the original meaning and should have a higher metric as a result of its supreme instruction-following capabilities. The issue with BERT-score arises from the inadequacy of underlying models in comprehending the semantics. Consequently, we use LLM(BGE-base Xiao et al. (2023) used)-score, the cosine similarity between LLM-induced embeddings to measure the semantic preservation. The experimental results in Table 1 demonstrate the effectiveness of BGE-base in assessing semantic preservation in complex style transfer tasks.

**Overall Metric: Geometric Mean**. We use the geometric mean of the style transfer accuracy and semantic preservation to assess the overall performance of models as well as the quality of the generated texts.

## 3 EXPERIMENTS

### 3.1 DATASETS

In this paper, we focus on two complex styles: character personalities in video games and narrative styles in long paragraphs. We establish a character personality benchmark and utilize three datasets from Zhu et al. (2022) for the latter. Dataset details can be found in Table 2.

Table 2: Statistics of the Genshin, Shakespeare and Luxun datasets. Note that Genshin* stands for the part for each character.

| Dataset | Train | Validation | Test |
|---|---|---|---|
| Genshin* | 9000 | 500 | 500 |
| Shakespeare | 2322 | 290 | 290 |
| LuXun | 1456 | 242 | 729 |
| JinYong | 1456 | 242 | 729 |

Table 3: Manual Evaluation results on the sampled test set of the Genshin dataset.

| Models | Style | Semantics | Overall |
|---|---|---|---|
| GST | 4.1 | 4.5 | 4.3 |
| TextSETTR | 3.7 | 3.6 | 3.6 |
| BTTS (T5-large) | 2.5 | 2.7 | 2.6 |
| Ours | 2.4 | 1.3 | 1.7 |
| ChatGPT | 1.1 | 2.3 | 1.5 |

In the **Genshin** dataset, we collect dialogues from the online open-world video game Genshin Impact, focusing on characters with distinct personalities: Fischl, Xiangling, Raiden Shogun, Nahida, Layla, and Wanderer. The parallel corpus of each character is created in the first stage of our Tiny-StyleWizard framework. Each parallel corpus consists of 10,000 samples. The **Shakespeare** dataset, designed for story author-style transfer Zhu et al. (2022), contains long input stories. To achieve style transfer to a Shakespearean narrative, we randomly select four examples from this dataset, resulting in a parallel corpus of 2,322 sentences. For the Chinese **LuXun/JinYong** dataset, we adopt a similar processing and construct a parallel corpus of 1456 sentences respectively.

### 3.2 BASELINES

To our knowledge, there is one study for the transfer of the personality of a game character under the few-shot setting Xu et al. (2023). Following this work, we adopt the following baselines:

**GST** Syed et al. (2020). This paper introduces an approach to generating author-stylized text using a denoising autoencoder (DAE) in a cascaded encoder-decoder framework. It leverages a pre-trained language model to rewrite input text into the targeted style without the need for parallel data.

**CP** Xu et al. (2020). This paper presents a new method using a variational autoencoder (VAE) and control variables to achieve controllable text representations. Two models, CP-G and CP-B, are considered, utilizing averaged GloVe and BERT as sentence representations, respectively. In our experiments, we only use CP-B due to the performance.

**TextSETTR** Riley et al. (2020). This model employs a large pre-trained language model to learn style representations and is capable of performing few-shot arbitrary style transfer.

**BTTS** Xu et al. (2023). This model improves Riley et al. (2020) by incorporating a contrastive loss to measure the similarity between input and extractor embeddings. The style extractor captures more precise style representations. This is the first work exploring the possibility for personality transfer.

For the story author style transfer, we consider the following baselines which are also adopted by the previous work Zhu et al. (2022) and itself:

**Style Transformer** Dai et al. (2019b). This model adds an extra style embedding and a discriminator to provide rewards for style transfer.

**StyleLM** Syed et al. (2020). This baseline produces the desired text based on the provided style token and a distorted version of the original text, following the model proposed by .

**Reverse Attention** Lee et al. (2021). It incorporates a reverse attention module into the final layer of the encoder, with the goal of removing the style information from the encoder's hidden states.

**StoryTrans** Zhu et al. (2022). It is the first long-story author-style transfer model and achieve state-of-the-art experimental results.

Table 6: Automatic Evaluation results on the test set of the Chinese LuXun and JinYong datasets.

| Models | LuXun | | | JinYong | | |
|---|---|---|---|---|---|---|
| | Style/% | Semantic/% | Overall/% | Style/% | Semantic/% | Overall/% |
| Style Transformer | 0.13 | **98.51** | 3.58 | 0.13 | 98.22 | 3.57 |
| StyleLM | 33.33 | 80.42 | 51.77 | 51.16 | 83.70 | 65.44 |
| Reverse Attention | 42.93 | 69.44 | 54.60 | 66.39 | 76.44 | 71.23 |
| StoryTrans | 74.35 | 73.33 | 73.83 | 90.67 | 68.82 | 78.99 |
| BART-base | 85.05 | 97.92 | 91.25 | 93.83 | **98.30** | 96.03 |
| BART-large | 92.18 | 97.14 | 94.62 | 97.53 | 97.89 | **97.71** |
| LongLM-small | 30.86 | 93.84 | 53.81 | 57.06 | 93.69 | 73.11 |
| LongLM-base | 73.53 | 93.43 | 82.88 | 86.97 | 94.50 | 90.65 |
| LongLM-large | 87.93 | 93.47 | 90.65 | 94.79 | 94.51 | 94.64 |
| ChatGPT | **97.67** | 94.54 | **96.09** | **99.45** | 95.66 | 97.53 |

## 3.3 AUTOMATIC EVALUATION RESULTS

### 3.3.1 RESULTS OF PERSONALITY TRANSFER

Table 4: Automatic Evaluation results on the test set of the **Genshin** dataset.

| Models | Style/% | Semantic/% | Overall/% |
|---|---|---|---|
| GST | 40.12 | 37.81 | 38.94 |
| CP-B | 20.51 | 38.85 | 28.22 |
| TextSETTR (T5-large) | 47.73 | 31.32 | 38.66 |
| BTTS (T5-large) | 50.24 | 37.36 | 43.32 |
| BART-base | 80.15 | **88.72** | 84.32 |
| BART-large | 88.34 | 88.47 | **88.40** |
| T5-base | 76.24 | 88.37 | 82.08 |
| T5-large | 82.27 | 88.54 | 85.34 |
| ChatGPT | **88.72** | 86.64 | 87.67 |

Table 5: Automatic Evaluation results on the test set of the **Shakespeare** dataset.

| Models | Style/% | Semantic/% | Overall% |
|---|---|---|---|
| Style Transformer | 0.01 | 93.84 | 0.96 |
| StyleLM | 3.44 | 89.30 | 17.52 |
| Reverse Attention | 0.01 | 92.92 | 0.96 |
| StoryTrans | 76.21 | 85.44 | 80.69 |
| BART-base | 96.90 | 93.88 | 95.37 |
| BART-large | **98.97** | 93.19 | **96.03** |
| T5-base | 88.97 | **94.59** | 91.73 |
| T5-large | 95.52 | 94.17 | 94.84 |
| ChatGPT | 94.03 | 93.74 | 93.88 |

Our experimental findings on the Genshin dataset (Table 4) demonstrate that small language models produced by the Tiny-StyleWizard framework outperform all other baselines in terms of style transfer accuracy and semantic preservation. Because baselines relying on few-shot learning struggle to infer implicit semantic transformation relationships, they cannot achieve considerable performance in semantic preservation as well as style transfer accuracy (Transfer on error aspects may lead to lower transfer accuracy).

Notably, our experimental results challenge the conventional wisdom that balancing style transfer accuracy and semantic preservation is difficult. Our model achieves optimal performance in both aspects, thanks to the high-quality parallel corpus generated by large language models. This corpus provides distinct stylized text and explicit semantic transformation relationships. The former can be satisfied with different piles of stylized corpus, while the latter is hard to collect and annotate. It is the exceptionally high-quality parallel corpus that enables us to achieve excellent results even with a relatively simple model architecture.

### 3.3.2 RESULTS OF NARRATIVE STYLE TRANSFER

In Table 5 and 6, we present an overview of the performance on the narrative style transfer task. Similar to our experiments on the Genshin dataset (Table 4), the small language models produced by our Tiny-StyleWizard framework achieve optimal performance in both style transfer accuracy and semantic preservation for English Shakespearean narrative style transfer as well as Chinese Jinyong-martial-art or LuXun-realism-novel narrative style transfer.

In contrast, the StoryTrans model, previously excelling in narrative style transfer, only achieves optimal style transfer accuracy while exhibiting subpar semantic preservation. Its two-stage generation process alters important entities (such as character names) into similar words from the targeted narrative style, leading to a lower level of semantic preservation. StyleTransformer, a purely attention-

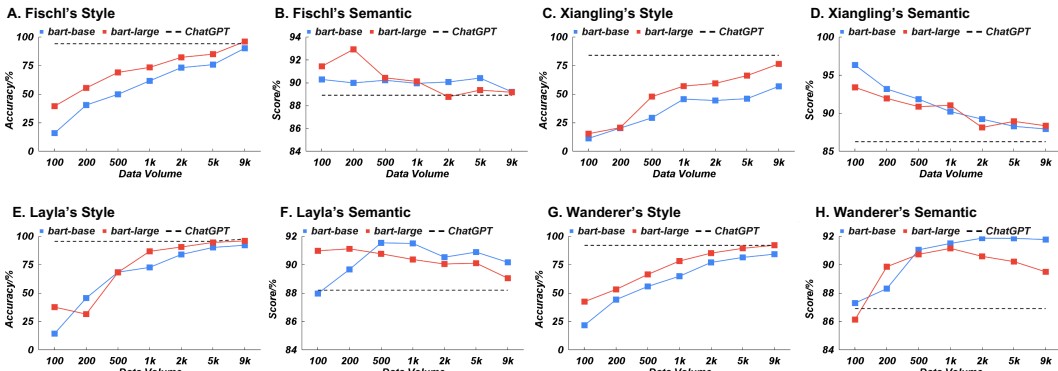

Figure 3: Style transfer accuracy and semantics preservation under different data volumes for various characters, Fischl in A and B, Xiangling in C and D, Layla in E and F, and Wanderer in G and H.

based model, struggles to remove style features and becomes similar to an autoencoder in narrative style transfer tasks. StyleLM performs better in content preservation due to denoising at the input stage but lacks disentanglement, resulting in lower style transfer accuracy. Reverse Attention excels in style accuracy but falls short in content preservation, indicating that reweighting hidden states allows for better style control rather than explicit word deletion.

## 3.4 MANUAL EVALUATION RESULTS

We employ human assessment alongside automated metrics to enhance our evaluation process. We randomly select 100 instances from each character within the Genshin datasets, which include the aforementioned six characters. We then request feedbacks from 10 participants who are tasked with ranking the models based on (1) the effectiveness of style transfer and (2) the preservation of semantic meaning. The highest-ranking model is awarded a rank of 1, while the lowest-ranking model receives a rank of 4. In Table 3, we can observe the average rankings) assigned to the four models according to the responses gathered from the participants. This data provides strong support for our assertion that our model excels in personality transfer tasks within the dataset.

## 3.5 ANALYSIS

### 3.5.1 EMERGENCE ANALYSIS FOR STYLE AND SEMANTICS

This section evaluates the performance of small language models produced by the Tiny-StyleWizard framework on complex style transfer tasks, which are trained on varying data volumes without optional ranking. We use different-sized models (BART-base and BART-large) and datasets (from 100 to 9k). The experimental result in Figure 3 revealing that style transfer is more challenging, requiring at least 5000 parallel sentences, and is significantly influenced by the model scale and dataset size. However, semantic preservation can be easily achievable with high-quality corpora and is less affected by model or dataset size, a trend also seen in simpler models like StyleTransformer.

### 3.5.2 LESS IS MORE

This section further investigates the impact of corpus filtering on the performance of small language models. We employ the metrics described in Section 2.3 to rank the quality of the generated corpora in descending order and extract different amounts of data, resulting in training corpora of varying sizes. We utilize these corpora to train personality transfer models for two characters, Xiangling and Raiden Shogun. These characters were chosen due to their relatively weaker performance metrics. The experimental results are depicted in the Figure 4. It is evident that the observed trends in the results differ significantly from Figure 3. With the inclusion of high-quality parallel corpora, the same model can quickly achieve optimal performance. Furthermore, the attained optimal performance surpasses the best one obtained via unfiltered corpora. However, as the data volume increases, the quality of the data gradually deteriorates, leading to a significant decline in model performance. It can be said that our well-trained models are contaminated by low-quality data during the continued

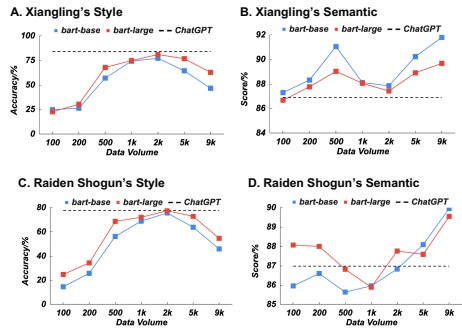

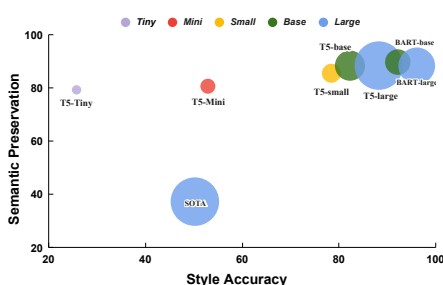

Figure 4: Performance under descent-rank high-quality data with different volumes for two characters, Xiangling and Raiden Shogun.

Figure 5: Bubble figure to indicate the minimum model scale for language models to emerge the capability for complex text style transfer.

training process. This phenomenon appears to resemble the concept of reverse curriculum learning, bearing similarities to the recently discovered concept of "ungrokking" Varma et al. (2023).

### 3.5.3 HOW SMALL CAN LANGUAGE MODELS TAKE EFFECTS?

we aim to investigate to what extent small-scale language models can still demonstrate the ability to transfer complex text styles. To address this, we fine-tuned two models, T5-tiny and T5-mini, on the character corpus of Fischl. The bubble chart (Figure 5) comparison shows T5-tiny excels in semantic preservation but lacks in style transfer, while T5-mini surpasses state-of-the-art models in both metrics yet underperforms compared to ChatGPT. This suggests that smaller language models (about 30M parameters) can effectively transfer complex styles when trained on suitable corpora.

### 3.5.4 TAKE A FURTHER STEP: DIVERSITY ANALYSIS

To measure the diversity of stylized texts generated by different models with varying data volumes, we use the Rouge score. This quantitative measure assesses the overlap of n-grams between a hypothesis text $H$ and a reference text $R$. The Rouge$k$ score is calculated based on the $k-$grams of $H$ and $R$, namely, $\text{Rouge}_{k,p}(H,R) = \frac{1}{|G_k(H)|} \sum_{t \in G_k(H)} 1_{t \in G_k(R)}$. The Rouge$k$ precision score measures how many $k$-grams in hypothesis text $H$ is included in that of reference text $R$. The final Rouge$k$ score (fmeasure) is given as $\text{Rouge}_k(H,R) = \frac{2 \times \text{Rouge}_{k,p}(H,R) \times \text{Rouge}_{k,p}(R,H)}{\text{Rouge}_{k,p}(H,R) + \text{Rouge}_{k,p}(R,H)}$. Inspired by Eldan & Li (2023), we perform the following experiments: We randomly pick 100 teacher-generated texts from the training dataset and generate corresponding responses via different trained small models. Let $t_1, ..., t_{100}$ be the response generated by a small model, and $t'_1, ..., t'_{100}$ be the responses generated by the teacher model, we are curious about the following two questions: **Q1**: What portion of the outputs produced by a small model can be found within the initial training responses? We apply the Equation $s_i = \text{Rouge}_{2,p}(t_i, t'_i)$ and illustrate the distribution in Figure 6. **Q2**: What is the degree of resemblance among the responses produced by a small model? To gauge this resemblance, we utilize the Equation $r_i = \max_{j \neq i} \text{Rouge}_2(t_i, t_j)$ and illustrate the distribution in Figure 7.

From the histograms depicted in Figure 6 and Figure 7, we can draw the following conclusions: 1) The stylized texts generated based on the training data often exhibit significant divergence from the initial responses. 2) There is a scarcity of common $k-$gram patterns between the generated responses and the dataset, resulting in distinctiveness among the generated texts. All in all, these findings strongly suggest that our models produce genuinely fresh and diverse texts rather than mere permutations of existing ones.

## 4 RELATED WORKS

Due to the scarcity of parallel corpora, most existing text style transfer methods primarily rely on unsupervised learning and train models on labeled but non-parallel corpus. Different architectures,

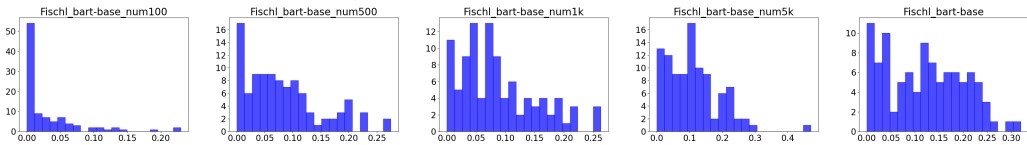

Figure 6: Rouge2 (precision) score between the generated texts obtained by our trained BART-base and the teacher model (ChatGPT) respectively. The dataset is personality transfer dataset for Fischl.

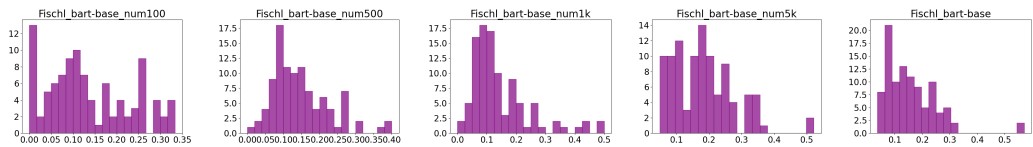

Figure 7: Maximum Rouge2 score (fmeasure) similarity between the 100 generated stylized texts for each model.

such as RNN Hochreiter & Schmidhuber (1997) and Transformers Vaswani et al. (2017), have been employed to learn semantic and style representations. However, these methods are limited by the availability of style labels, which are often lacking for many desired attributes in real-world applications. Additionally, the performance of text style transfer models trained on non-parallel parallel corpus is still bounded by the parallel counterparts.

A few-shot approach was recently introduced by Riley et al. (2020), which trains on massive amounts of unlabeled corpora and requires only a few samples during inference, even for newly-appearing style transfer. While this approach eliminates the need for labeled training samples, it often struggles when it comes to complex style transfer tasks Xu et al. (2023). The advancements in large language models (LLMs) have demonstrated intriguing abilities for generating fluent, meaningful, satisfactory texts across a myriad of domains OpenAI (2023). Some studies have explored the use of LLMs for zero-shot text style transfer Reif et al. (2021), where a natural language transfer instruction is only needed instead of fine-tuning or target style exemplars. For instance, Reif et al. (2021) proposes an augmented zero-shot learning method for arbitrary text style transfer based on LLMs. This method demonstrates superior performance on standard style transfer tasks as well as more creative ones. Furthermore, this work showcases the ability to handle multiple styles simultaneously and generate diverse outputs. Pu and Demberg Pu & Demberg (2023) demonstrated that in text style transfer, ChatGPT surpassed the previous state-of-the-art models by achieving better results in few-shot learning, as indicated by the higher BLEU score. However, when it comes to controlling the formality, ChatGPT still shows noticeable distinctions from human behavior. Our work takes a further step, exploring the emergent ability of text style transfer in small-scale models.

## 5 CONCLUSION

In conclusion, this paper has presented Tiny-StyleWizard, a novel framework designed to address the challenges of complex style transfer. By generating a specialized dataset with a large language model and subsequently fine-tuning a small language model, we have demonstrated the feasibility of complex style transfer. The innovative evaluation protocol has allowed us to rank the quality of the generated dataset and measure the performance of various models reliably and effectively. Extensive experiments have revealed that small language models can generate stylized text comparable to ChatGPT, and even smaller models like T5-mini can surpass state-of-the-art models. Our research into the efficient construction of the training corpus has led to the "less is more" and "ungrokking" observations, highlighting the paramount importance of data quality. This work not only contributes to the field of complex style transfer but also provides valuable insights into the role of data quality and model size in other tasks. Future work will continue to refine our framework and explore its potential applications in various natural language processing tasks.

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
