# OpenReview forum: "Tiny-StyleWizard: Unleashing the Potential of Small Language Models in Complex Style Transfer"
_ICLR.cc/2024/Conference — Submitted to ICLR 2024_

### Official Review · Reviewer_B8Me · 2023-10-16

**Soundness:** 3 good
**Presentation:** 1 poor
**Contribution:** 2 fair
**Rating:** 3
**Confidence:** 3

**Summary:**

In this paper, the authors presented TinyStyleWizard, which is a framework for generating specialized dataset for complex style transfers, and then fine-tune small language models on the datasets. The authors also devised some novel evaluation metrics for the tasks. The authors compared their fine-tuned models against baselines on their newly generated datasets, and achieved good results. The authors also performed additional analysis on the relationships between performance and training data volume, as well as generation diversity.

**Strengths:**

1. The authors presented a new benchmarks that includes several new datasets for complex style transfers. These datasets could be beneficial for future research in text style transfers.

2. The research focused on enabling small language models to perform text style transfer tasks, which is really interesting because even though large language models can easily perform the same tasks, small language models are much cheaper and less constrained to finetune and use, and how we can utilize large language models to "teach" small language models remains challenging.

**Weaknesses:**

The paper claims to study "the potential of small language models in complex style transfer", but it did not present a clear definition of "small language model" or "complex style transfer". It is unclear how small the model needs to be to be considered a "small LM". It is also unclear which transfers can be considered "complex". I believe the authors need to provide a concrete definition for both terms (such as a clear size boundary for "small LM" and a clear linguistic definition of "complex style transfer"). The paper currently uses both terms vaguely, which makes it hard to accurately assess the significance of the contribution, as there already exists many parallel corpuses of style transfer that could be considered "complex", such as formality transfer (GYAFC[1]) or composition of multiple fine-grained text style transfer (StylePTB[2]). The paper could also benefit from evaluating TinyStyleWizard models on these existing and well-defined parallel corpuses to show the generalizability of the proposed methods.

There is no details on how the TinyStyleWizard models are fine-tuned from their pre-trained state.

In addition, there are a few serious issues in the presentation of the paper that significantly increases difficulty of reading, including:

(1) The in-text citation style seems to be wrong, i.e. there are no parentheses around each in-text citation, which makes the text much harder to read, especially in the introduction and related work sections where there are a lot of citations.

(2) The Tables are very difficult to comprehend and often there is no clear takeaway messages. See Questions for details.

(3) The ordering of the tables are quite strange. Table 3 is referred in later context than Table 4,5,6. Table 6 appears in the paper above Tables 4 and 5.

References:
[1] https://aclanthology.org/N18-1012/
[2] https://arxiv.org/abs/2104.05196

**Questions:**

1. I found most of the tables in the paper difficult to understand.

(a) For Table 1, it is unclear to me what the takeaway message should be. Is it used to show the quality of the corpus you collected? Or is it showing that your metric is better? In either case, it is hard to draw conclusions from the bolded numbers in the table.

(b) For Table 4,5,6, why are the baselines different in these settings?

(c) For Table 3, which TinyWizard model is "ours"? (i.e. which row from Table 4 corresponds to "ours" in Table 3?) You should also bold the best numbers from Table 3, since you bolded best numbers from all other tables.

2. Why do we want "diversity" (sec 3.5.4) in style transfer tasks? If the style is precisely defined and the semantic preservation is perfect, then all results should be similar and a "diverse" result would either mean poor style accuracy in some transfers or poor semantic preservation in some transfers.

---

> ### Author Response · Authors · 2023-11-21
>
> Firstly, we want to sincerely thank you for the time and effort you've put into reviewing our paper and offering insightful feedback. We have thoroughly reviewed each point you raised and have responded to the key issues in the sections below. We trust that our replies will resolve any uncertainties and further highlight the significance and contributions of our research.
>
> * The small language models we refer to are those with fewer than 100 million parameters, such as T5-tiny (31M) and T5-small (60M). As can be seen from Figure 5 in the paper, both of these small language models outperform the existing best models in terms of style transfer accuracy and semantic preservation.
> * The complex text style transfer we claim originates from the BTTS paper[1]. In that work, they provide a preliminary definition of complex text style, which refers to styles that are difficult for non-experts to discern and categorize. Such styles might include personality, domain-specific jargon, or other highly specialized terminology. The complexity of these styles makes it challenging to use crowdsourcing for labeling, as only experts in the relevant fields can accurately differentiate between styles. Therefore, the challenge in complex text style transfer is to develop effective models capable of capturing and transferring these nuanced styles without relying heavily on labeled data.
> * Details on how the models were fine-tuned: we fine-tuned the pretrained language models (BART, T5, LongLM etc.) with the huggingface coding framework on single A100 GPU. The initial learning rate is 2e-5. The batch size is 4.
> * We will correct the citation issues in subsequent versions;
> * We will also modify the layout of the tables according to your suggestions.
> * Reading Table 1, the main takeaways are two-fold: First, using a few-shot classifier based on ChatGPT to calculate the accuracy of style transfer is effective, and it performs better than the model fine-tuned on the training set using DeBERTa. Second, using LLM-score to assess the degree of semantic preservation in style transfer is more reliable than other metrics. Our reasoning is that ChatGPT, trained on massive corpora and reinforced through feedback alignment, naturally preserves the semantics of the original text in its stylized output. A good assessment metric should, therefore, yield a high similarity when evaluating the original text against the text generated by ChatGPT.
> * Regarding the differences in baselines: Table 4 shows the experimental results on the Genshin dataset, while Table 5 presents the results on the Shakespeare dataset. The former involves shorter text lengths (under 30 words), and the latter longer text lengths (average length of about 70 words). Both represent complex styles and have some foundational work. We established the baselines based on their respective foundational works.
> * In manual evaluations, we used the fine-tuned BART-large model. We will bold the performance of the best models in all tables.
> * As for diversity you mentioned, our game designers hope to express similar semantics within the same style using diverse wordings. According to their rich experience, players do not appreciate encountering repetitive and identical text throughout the game, as it can lead to aesthetic fatigue and diminish their gaming experience. Moreover, we hope that the stylized content viewed by different players will also vary in terms of wrod choice. Therefore, we believe that this metric is extremely important for the gaming business. Thus, we take this diversity metric into accout as well as the conventional style transfer accuracy and semantic preservation.
>
> References
>
> [1] Specializing Small Language Models towards Complex Style Transfer via Latent Attribute Pre-Training

---

### Official Review · Reviewer_Yb3s · 2023-10-31

**Soundness:** 3 good
**Presentation:** 2 fair
**Contribution:** 2 fair
**Rating:** 3
**Confidence:** 5

**Summary:**

The authors propose approaches using “small” language models to address “complex” style transfer, i.e. addressing style transfer in the context of more complex styles associated with unique personalities and long form narratives. They propose a chatGPT based evaluation, which addressing weaknesses with existing automatic evaluations (e.g. SBERT), and also use chatGPT as a data augmentation tool to create parallel datasets for fine-tuning. The demonstrate that their fine-tuned approaches operate on par with chatGPT across several different evaluations on two datasets. They also conduct studies on the amount of data and quality of finetuning data, in line with prior work, they find that quality is crucial.

**Strengths:**

The authors cover a significant amount of related work, and compare their approaches to a good set of benchmarks (though they do not compare consistently across datasets). They also conduct good studies on data quality and generation diversity. The research questions are well outlined and motivated in the introduction, however there are significant weaknesses in addressing them (see below).

**Weaknesses:**

* Multiple instances of improper grammar (ex. Sentence 2 of section 2 “...it generates”) and unusual wording. Difficult to understand at times with grammar issues.
  * “unleash the capability of these models”
  * “Small language models specialized for the magician capabilities of style transfer”
  * 3.5.3 title typo “How small can language models take effects”. The first word in this paragraph should also be capitalized.
* Re data generation: figure 2 and section 2.1 indicate chatGPT is used in conjunction with a instruction based prompt to generate synthetic data, but there are no details on what exactly the prompting strategy is, or what any of the results looks like
* Details lacking on chatGPT based evaluation metric - no description of how this actually works. This is especially important as the authors highlight the proposed metric as one of their 3 novel contributions. In Table 1, chatGPT is reported but there is no corresponding description of how it is used, the same table also lists ‘fine-tuned’ but doesn’t not indicate what is fine-tuned.

Minor issues
* In section 2.3, “However, our analysis has identified..” it would be nice to have a forward reference to the experiment here
* Table 1 can be moved to a later page, closer to when it’s referenced
* Table 3: There is no metric for the values in the table, not sure what is being reported
* Consider adding https://arxiv.org/pdf/2010.05700.pdf to related work, as there are other ways to getting around a lack of parallel corpora.

**Questions:**

* In the human evaluations (section 3.4) are you certain that the evaluators are not conflating strong fluency for style matching? Especially for style associated with a particular character, it may be hard to judge which is best (if they aren’t familiar with the characters). This may be an explanation for why chatGPT performs well.

* Why aren’t the same baselines used in Tables 4 and 5?

---

> ### Author Response · Authors · 2023-11-20
>
> At the outset, we wish to extend our heartfelt thanks for the time and effort you invested in reviewing our paper and offering constructive feedback. We have diligently examined each of your observations and have tackled the primary issues in the subsequent sections. Our aim is that our detailed responses will clear up any uncertainties and more effectively showcase the importance and impact of our study.
>
> - We will revise the terms that are difficult to understand. Some modifications are listed as follows:
>
>   - unleash the capability of these models -> fully utilize the potential of these models
>   - Small language models specialized for the magician capabilities of style transfer -> Small language models specialized for style transfer
>
> - We will present the prompts we used and the related results here. We used ChatGPT as a few-shot classifier for categorizing text styles. The prompt we used is shown beneath. [Prompt text here]. The term 'fine-tuned model' refers to the model we obtained by fine-tuning DeBERTa on the training set containing a variety of different style corpora. In our work, the training dataset is the training part of genshin, Shakespeare, LuXun and JinYong.
>
>   The prompt used for evaluating the style transfer accuracy:
>
>   ```shell
>   ${CHARACTER_PROFILE}
>
>   You are an excellent style classifier, please determine whether the style presented in the input sentence is consistent with ${YOUR_CHARACTER}'s speaking style. Output either `yes` (consistent) or `no` (inconsistent).
>
>   ${DEMO}
>
>   input sentence: ${text}
>   label:
>   ```
>
> - We will adjust the layout of the figures and references in the paper based on your suggestions.
>
> - We invited 10 designers from the gaming industry to evaluate the texts generated by different models. These designers have played multiple open-world games, including 'Genshin Impact.' They are capable of discerning and judging the quality of the stylized texts generated by the models.
>
> - Regarding the differences in baselines: Table 4 shows the experimental results on the Genshin dataset, while Table 5 presents the results on the Shakespeare dataset. The former has shorter text lengths (under 30 words), and the latter has longer text lengths (average length of about 70 words). Both represent complex styles and have some preliminary work. We set up the baselines based on their respective preliminary works.

---

### Official Review · Reviewer_KX34 · 2023-11-01

**Soundness:** 3 good
**Presentation:** 3 good
**Contribution:** 2 fair
**Rating:** 6
**Confidence:** 4

**Summary:**

The paper presents a technique for complex style transfer using the following steps:
- Choose a style
- Create a small, high quality training dataset (~O(10k)) of that style using an LLM
- Use that dataset to finetune a much smaller model which is ultimately used.
They also propose an evaluation to rank the quality of the generated examples to choose which to use for finetuning, and to evaluate style transfer across a few dimensions.

**Strengths:**

- It's great to see approaches like this which recognize that LLMs are not tractable in many situations. Additionally, it's interesting to see how LLMs can be leveraged for small, impactful parts of the pipeline (e.g., creating a small and high quality dataset)
- The paper is well-written and straightforward; I enjoyed the descriptions of complex style transfer being "magician capabilities" :)

**Weaknesses:**

- While the end-to-end system is an interesting use case, I'm not sure this is particularly novel overall. We know that LLMs can create small, high quality datasets, and that those datasets can be used to train smaller models.
- Using a different model's embedding space to calculate BERT-score doesn't seem significant enough to include as a contribution
- Nit: use \citep vs \citet for clarity (https://www.reddit.com/r/LaTeX/comments/5g9kn1/whats_the_difference_between_cite_citep_and_citep/)
- "...challenge the conventional wisdom that balancing style transfer accuracy and semantic preservation is difficult" this seems like a strong claim. In my mind, this tradeoff has not been an issue for LLM-based style transfer methods.

**Questions:**

- Do you have the prompts that you used for the style transfer accuracy classifiers?
- Also for the style transfer itself (in Table 6)?

---

> ### Author Response · Authors · 2023-11-20
>
> We greatly appreciate the time and effort you dedicated to reviewing our paper and offering insightful feedback. We have thoughtfully reviewed each of your remarks and have responded to the key concerns in the sections below. It is our hope that these clarifications will resolve any outstanding questions and further highlight the significance and contributions of our research.
>
> * Our research not only demonstrates that large language models can generate high-quality datasets for training smaller models with comparable performance in complex style transfer tasks, but also investigates the minimal size of language models required to effectively execute complex style transfer. Furthermore, it explores how different dimensions of complex style—specifically, the accuracy of style transfer and the degree of semantic preservation—place varying demands on the size of the language model.
>
> * The calculation method of LLM-score is distinct from that of BERT-score. LLM-score assesses the similarity between model-generated text and reference text at the level of the whole sentence (cosine similarity). In contrast, BERT-score requires calculating the precision and recall at the word/token level for the generated text in comparison to the reference text. We believe that complex text styles are manifested across entire sentences, rather than being uniformly distributed across each word.
>
> * Thank you for pointing out the issue with inappropriate citation. We will address this in subsequent versions of our paper.
>
> * This traditional perspective has been discussed in the previous paper[1]. At that time, large language models, such as ChatGPT, had not yet achieved remarkable success in various natural language processing tasks. Following the emergence of ChatGPT, our view, similar to yours, is that introducing high-quality parallel corpora generated by large language models can break this trade-off. Therefore, we conducted this study. The experimental results in our paper demonstrate the validity of our idea.
>
> * Our prompt for style transfer accuracy classifier:
>
>   ```shell
>   ${CHARACTER_PROFILE}
>
>   You are an excellent style classifier, please determine whether the style presented in the input sentence is consistent with ${YOUR_CHARACTER}'s speaking style. Output either `yes` (consistent) or `no` (inconsistent).
>
>   ${DEMO}
>
>   input sentence: ${text}
>   label:
>   ```
>
> * Our prompt for style transfer in Table 6:
>
>   The prompt to generate parallel corpus for LuXun-style transfer is shown beneath. The initial parallel sample(s) used for demonstration is obtained by zero-shot prompting. The prompt to generate parallel corpus for JinYong-style transfer is similar.
>
>   ```shell
>   给定一段童话文本，在保持原文语义不变的情况下，将其转换为带有鲁迅风格的文本。
>
>   ${DEMO}
>
>   童话风格的文本：${text}
>   鲁迅风格的文本：
>
>   Given a piece of fairy tale text, it is transformed into a text with the style of Lu Xun while maintaining the original meaning.
>
>   ${DEMO}
>
>   Fairy-tale style text: ${text}
>   LuXun-style text:
>   ```
>
> [1] Storytrans: Non-parallel story author-style transfer with discourse representations and content enhancing

---

### Official Review · Reviewer_Lzy3 · 2023-11-01

**Soundness:** 2 fair
**Presentation:** 3 good
**Contribution:** 3 good
**Rating:** 3
**Confidence:** 5

**Summary:**

The paper presents a simple but effective method of text style transfer (TST), which first generates a parallel corpus of the target style with few-shot prompts, and then train a seq2seq model over the parallel corpus. The experiments confirm the effectiveness of the method in several benchmarks.

**Strengths:**

**Clarity and Significance**

- The paper is well-written and easy to follow. The improvement in small LM TST performance is significant, although such gain should be attributed to the generated parallel datasets.
- The method is relatively easy and effective.

**Weaknesses:**

**Evaluation**

My main concern about this work is the evaluation: afaik there are several other strong TST baselines that also leverage parallel datasets (either synthetic or mined from unsupervised ground truth) such as TSF-DelRetGen [1], IMaT [2], STRAP [3], LaMer [4] which are missed in the current work. IMaT proposed to use translation to generate parallel/matching datasets, STRAP framed the TST task as a paraphrase generation tasks, and LaMer mined roughly parallel data from unparallel TST datasets, which I believe all have similar ideas as yours. Another baseline is [5], which finds directly asking LLMs about target style text with few-shot is enough to achieve SotA performance --- if that is the case, what's the benefit of distilling the data into a small TST model (and maybe less generalized with limited performance)?

The human evaluation is crucially ignored, which I believe is the most important metric we should care for TST tasks (as pointed out in [3], automated metrics such as BLEU/ROUGE can be easily gamed).

In general, the evaluation is not very convincing to me as the missing strong baselines and human evaluation.




[1] Transforming Delete, Retrieve, Generate Approach for Controlled Text Style Transfer

[2] IMaT: Unsupervised Text Attribute Transfer via Iterative Matching and Translation

[3] Reformulating Unsupervised Style Transfer as Paraphrase Generation

[4] Non-Parallel Text Style Transfer with Self-Parallel Supervision

[5] A Recipe for Arbitrary Text Style Transfer with Large Language Models

**Questions:**

Could you discuss the difference/contribution of your method compared with the above-mentioned work in the revised version?

---

> ### Author Response · Authors · 2023-11-20
>
> First and foremost, we would like to express our sincere gratitude for your time and effort in reviewing our paper and providing valuable feedback. We have carefully considered each of your comments and addressed the main concerns in the following sections. We hope that our responses will clarify any issues and further demonstrate the value and contribution of our work.
>
> In our initial experiments, we employed the GST model[1] as a representative of strong supervised baselines which leverage parallel dataset (either synthetic or mined from unsupervised ground truth). The results demonstrated that the performance of the GST model on the Genshin dataset was inferior to that of our proposed method. Similar experimental outcomes have been observed in previous studies[2]. The last paper[3] you mentioned directly employs a large language model combined with specialized prompting techniques for text style transfer. We believe that this approach is fundamentally analogous to using ChatGPT for text style transfer, which is why we did not include it as a baseline in our study. Our experimental findings indicate that our method achieves superior results in text style transfer tasks compared to ChatGPT.
>
> The primary reasons for employing small language models in complex style transfer are attributable to the proprietary nature of closed-source large language models such as OpenAI's ChatGPT, the substantial deployment costs associated with open-source large language models as well as the significant inference latency.
>
> We also conducted a manual evaluation on the Genshin dataset, with the results presented in Table 3. The participants in the manual evaluation were professionals from the gaming industry, specifically game designers. These designers have experience playing multiple open-world games, including Genshin. They possess the expertise to accurately identify whether the texts generated by the model align with the characters' established settings and personalities.
>
> [1] Transforming Delete, Retrieve, Generate Approach for Controlled Text Style Transfer
>
> [2] Specializing Small Language Models towards Complex Style Transfer via Latent Attribute Pre-Training
>
> [3] A Recipe for Arbitrary Text Style Transfer with Large Language Models

---

### Meta-Review · Area_Chair_NBdB · 2023-12-13

**Metareview:**

This paper presents an empirical study of distillation for textual style transfer. An LLM is used to generate a parallel supervised corpus for a new style transfer task. Then a smaller LM is fine-tuned to perform style transfer directly. Reviewers pointed out the potential value of the newly generated data itself, which could be used as a new benchmark. Some reviewers valued the focus on smaller, more efficient solutions to style transfer given that LLMs are not always practical. However, reviewers generally indicated that they did not consider this paper ready for acceptance in its current form. Specifically, reviewers mentioned issues with the organization of experimental tables, inconsistency in which baselines were compared with on which tasks, missing baselines from the style transfer literature, missing details on how fine-tuning was performed, and limited empirical takeaways given that distillation has already been shown to be effective in related tasks.

**Justification For Why Not Higher Score:**

While this paper has promise (particularly the empirical demonstration of highly performant, small LMs for style transfer), there were too many issues brought up by reviewers for a single review period, e.g. missing details, questions about baseline comparisons, and limited experimental takeaways.

**Justification For Why Not Lower Score:**

N/A

---

### Decision · Program_Chairs · 2024-01-16

Reject